# Multi-Objective Online Learning

## Abstract

This paper presents a systematic study of multi-objective online learning. We first formulate the framework of Multi-Objective Online Convex Optimization, which encompasses two novel multi-objective regret definitions. The regret definitions build upon an equivalent transformation of the multi-objective dynamic regret based on the commonly used Pareto suboptimality gap metric in zero-order multi-objective bandits, making it amenable to be optimized via first-order iterative methods. To motivate the algorithm design, we give an explicit example in which equipping OMD with the vanilla min-norm solver for gradient composition will incur a linear regret, which shows that only regularizing the iterates, as in single-objective online learning, is not enough to guarantee sublinear regrets in the multi-objective setting. To resolve this issue, we propose a novel min-regularized-norm solver that regularizes the composite weights. Combining min-regularized-norm with OMD results in the Doubly Regularized Online Mirror Multiple Descent algorithm. We further derive both the static and dynamic regret bounds for the proposed algorithm, each of which matches the corresponding optimal bound in the single-objective setting. Extensive experiments on both simulation and real-world datasets verify the effectiveness of the proposed algorithm.

## 1 Introduction

Traditional optimization methods for machine learning are usually designed to optimize a single objective. However, in many real-world applications, we are often required to optimize multiple correlated objectives concurrently. For example, in autonomous driving [12, 20], the self-driving vehicles need to solve multiple tasks such as self-localization and object identification at the same time. In online advertising [21, 22], advertisers need to determine the exposure of items to different users to maximize both the Click-Through Rate (CTR) and the Post-Click Conversion Rate (CVR). In many multi-objective scenarios, the objectives may conflict with each other [15]. Hence, there may not exist any single solution that optimizes all the objectives simultaneously. For example, in online advertising, merely optimizing CTR or CVR will degrade the performance of the other [21, 22].

Multi-objective optimization (MOO) [23, 6] is concerned with optimizing multiple conflicting objectives simultaneously. It seeks Pareto optimality, where no single objective can be improved without hurting the performance of the others. Many different methods for MOO have been proposed, including evolutionary methods [26, 39], scalarization methods [9], and gradient-based iterative methods [7]. Recently, the Multiple Gradient Descent Algorithm (MGDA) and its variants have been introduced to the training of multi-task deep neural networks and achieved great empirical success [29], making them regain a significant amount of research interest [17, 33, 18]. These methods compute a composite gradient based on the gradient information of all the individual objectives and then apply the composite gradient to update the model parameters. The composite weights are determined by a min-norm solver [7] which yields a common descent direction of all the objectives.

However, compared to the increasingly wide application prospect, the gradient-based iterative algorithms are relatively understudied, especially for the online learning setting. Multi-objective online learning is of essential importance due to reasons in two folds. First, due to the data explosion in many real-world scenarios such as web applications, making in-time predictions requires performing online learning. Second, the theoretical investigation of multi-objective online learning will lay a solid foundation for the design of new optimizers for multi-task deep neural networks. This is analogous to the single-objective setting, where nearly all the optimizers for training DNNs are initially analyzed in the online setting, such as AdaGrad [8], Adam [16], and AMSGrad [28].

In this paper, we give a systematic study of multi-objective online learning. To begin with, we formulate the framework of Multi-Objective Online Convex Optimization (MO-OCO). The first major challenge is the lack of regret definitions in the multi-objective setting. To tackle this challenge, we need appropriate discrepancy metrics that can be used in the regret definitions, which evaluate the gap between any two vector losses by producing scalar values. Intuitively, the Pareto suboptimality gap (PSG) metric, which is frequently used in zero-order multi-objective bandits [30, 19], is a very promising candidate. It can yield scalarized distances from any vector loss to a given comparator set. We can thus define the multi-objective regret by simply plugging in PSG as the discrepancy metric. However, as a metric designed purely from the geometric view, PSG is intrinsically difficult to be optimized directly via gradient-based iterative methods. To resolve this problem, for the PSG-based multi-objective dynamic regret, we derive its equivalent unconstrained max-min form via a highly non-trivial transformation. This form is intuitive to the design of first-order multi-objective online algorithms, indicating that we should select a convex combination of the gradients at each round. Unfortunately, for the PSG-based static variant, such an equivalence does not exist. To remedy this issue, we make extensions of the dynamic variant by fixing the comparator set and the composite weights, which yields an appropriate definition of the multi-objective static regret.

Based on the MO-OCO framework, we develop a novel multi-objective online algorithm termed Doubly Regularized Online Mirror Multiple Descent. The key module of the algorithm is the gradient composition scheme, which calculates a composite gradient in the form of a convex combination of the gradients of all objectives. Intuitively, the most direct way to determine the composite weights is to apply the min-norm solver [7] commonly used in offline multi-objective optimization. However, directly applying min-norm is not workable in the online setting. Specifically, the composite weights in min-norm are only determined by the gradients at the current round. In the online setting, since the gradients can be adversarial, they may result in undesired composite weights, further producing a composite gradient that reversely optimizes the loss. To rigorously verify this point, we give a showcase in which equipping OMD with vanilla min-norm even incurs a linear regret, showing that only regularizing the iterate, as in OMD, is not enough to guarantee sublinear regrets in the multi-objective setting. To fix this issue, we devise a novel min-regularized-norm solver with an explicit regularization on composite weights. Equipping it with OMD results in our proposed algorithm.

We then conduct the theoretical analysis for our proposed algorithm. We derive a multi-objective static regret bound $O(\sqrt{T})$ and a multi-objective dynamic regret bound $O(V_T^{1/3}T^{2/3})$ for DR-OMMD. Both bounds match the optimal bounds in the single-objective setting [11, 34]. Our analysis also shows that DR-OMMD attains a lower regret than linearization with fixed composite weights.

To evaluate the effectiveness of DR-OMMD, we conduct extensive experiments on both simulation datasets and real-world datasets. We first elaborate simulation experiments, in which we find that DR-OMMD attains lower regret than vanilla min-norm and linearization, which verifies the superiority of the min-regularized-norm solver. We then realize adaptive regularization via multi-objective optimization on real-world datasets, and find that adaptive regularization with DR-OMMD significantly outperforms fixed regularization with linearization.

In summary, in this paper, we give the first systematic study of multi-objective online learning, which encompasses a novel framework, a new algorithm, and corresponding non-trivial theoretical analysis. We believe that this work paves the way for future research on more advanced multiple-objective optimization algorithms, which may inspire the design of new optimizers for multi-task deep learning.

## 2 Preliminaries

In this section, we briefly review the necessary background knowledge of online convex optimization and multi-objective optimization.

## 2.1 Online Convex Optimization

**Online Convex Optimization (OCO)** [38, 11] is the most commonly adopted framework for designing online learning algorithms. It can be viewed as a structured repeated game between a learner and an adversary. At each round $t \in \{1, \ldots, T\}$, the learner is required to generate a decision $x_t$ from a convex compact set $\mathcal{X} \subset \mathbb{R}^n$. Then the adversary replies the learner with a convex function $f_t : \mathcal{X} \to \mathbb{R}$ and the learner suffers the loss $f_t(x_t)$. The goal of the learner is to minimize the regret with respect to the best fixed decision in hindsight, i.e.,

$$R_S(T) = \sum\nolimits_{t=1}^{T} f_t(x_t) - \min_{x^* \in \mathcal{X}} \sum\nolimits_{t=1}^{T} f_t(x^*).$$

Note that the above regret is the **static regret** [10], which compares the learner's cumulative loss with that of a fixed decision. There is another version of regret, namely the **dynamic regret** [10, 34], which compares the learner's cumulative loss with that of a sequence of local optimal decisions, i.e.,

$$R_D(T) = \sum\nolimits_{t=1}^{T} f_t(x_t) - \sum\nolimits_{t=1}^{T} \min_{x_t^* \in \mathcal{X}} f_t(x_t^*).$$

Any meaningful regret is required to be sublinear in $T$, i.e., $\lim_{T \to \infty} R_{S/D}(T)/T = 0$, which implies that when $T$ is large enough, the learner can perform as well as the best fixed decision in hindsight (for static regret) or the local optimal decision at each round (for dynamic regret).

**Online Mirror Descent (OMD)** [11] is a classic first-order online learning algorithm. At each round $t \in \{1, \ldots, T\}$, OMD yields its decision using the following formula

$$x_{t+1} = \arg\min_{x \in \mathcal{X}} \eta \langle \nabla f_t(x_t), x \rangle + B_R(x, x_t),$$

where $\eta$ is the step size, $R : \mathcal{X} \to \mathbb{R}$ is the regularization function, and $B_R(x, x') = R(x) - R(x') - \langle \nabla R(x'), x - x' \rangle$ is the Bregman divergence induced from $R$. As a meta-algorithm, by instantiating different regularization functions, OMD can induce two important algorithms, i.e., Online Gradient Descent [38, 13] and Online Exponentiated Gradient [11].

## 2.2 Multi-Objective Optimization

**Multiple-objective optimization (MOO)** is concerned with solving the problems of optimizing multiple objectives simultaneously [39, 29]. In general, since different objectives may conflict with each other, there is no single solution that can optimize all the objectives at the same time. Instead, MOO seeks to find solutions that achieve Pareto optimality. Next, we exposit Pareto optimality and related definitions more formally using a vector-valued loss $H = (h^1, \ldots, h^m)^\top$ as objectives, where $m \geq 2$ and $h^i : \mathcal{K} \to \mathbb{R}, i \in \{1, \ldots, m\}, \mathcal{K} \subset \mathbb{R}$, is the $i$-th loss function.

**Definition 2.1** (**Pareto optimality**). **(a)** For any two solutions $x, x' \in \mathcal{K}$, we say that $x$ dominates $x'$, denoted as $x \prec x'$ or $x' \succ x$, if $h^i(x) \leq h^i(x')$ for all $i$, and there exists one $i$ such that $h^i(x) < h^i(x')$; otherwise, we say that $x$ does not dominate $x'$, denoted as $x \not\prec x'$ or $x' \not\succ x$.
**(b)** A solution $x^* \in \mathcal{K}$ is called Pareto optimal if it is not dominated by any other solution in $\mathcal{K}$.

There may exist multiple Pareto optimal solutions. For example, it is easy to show that the optimizer of any single objective, i.e., $x_i^* \in \arg\min_{x \in \mathcal{K}} h^i(x), i \in \{1, \ldots, m\}$, is Pareto optimal. Different Pareto optimal solutions reflect different trade-offs among the objectives [17].

**Definition 2.2** (**Pareto front**). **(a)** All Pareto optimal solutions form the Pareto set $\mathcal{P}_\mathcal{K}(H)$.
**(b)** The image of $\mathcal{P}_\mathcal{K}(H)$ constitutes the Pareto front, denoted as $\mathcal{P}(H) = \{H(x) \mid x \in \mathcal{P}_\mathcal{K}(H)\}$.

Now that we have established the notion of optimality in MOO, we proceed to introduce the metrics that measure the discrepancy of an arbitrary solution $x \in \mathcal{K}$ from being optimal. Recall that, in the single-objective setting with merely one loss function $h : \mathcal{Q} \to \mathbb{R}$, where $\mathcal{Q} \subset \mathbb{R}$, for any $z \in \mathcal{Q}$, the loss gap $h(z) - \min_{z'' \in \mathcal{Q}} h(z'')$ is directly the discrepancy measure. However, in MOO with more than one loss, for any $x \in \mathcal{K}$, the loss gap $H(x) - H(x'')$, where $x'' \in \mathcal{P}_\mathcal{K}(H)$, is a vector. Intuitively, the desired discrepancy metric shall scalarize the vector-valued loss gap and yield the value $0$ for any Pareto optimal solution. In general, there are two commonly used discrepancy metrics in MOO, i.e. Pareto suboptimality gap (PSG) [30] and Hypervolume (HV) [4]. As HV is a volume-based metric, it is more difficult to optimize or analyze via iterative algorithms [36]. Hence in this paper, we adopt PSG, which has been extensively used in multi-objective bandits [30, 19].

**Definition 2.3 (Pareto suboptimality gap).** For any $x \in \mathcal{K}$, the Pareto suboptimality gap to a given comparator set $\mathcal{K}^* \subset \mathcal{K}$, denoted as $\Delta(x; \mathcal{K}^*, H)$, is defined as the minimal scalar $\epsilon \geq 0$ that needs to be subtracted from all entries of $H(x)$, such that $H(x) - \epsilon \mathbf{1}$ is not dominated by any point in $\mathcal{K}^*$, where $\mathbf{1}$ denotes the all-one vector in $\mathbb{R}^m$, i.e.,[1]

$$\Delta(x; \mathcal{K}^*, H) = \inf_{\epsilon \geq 0} \epsilon, \quad \text{s.t. } \forall x'' \in \mathcal{K}^*, \exists i \in \{1, \ldots, m\}, h^i(x) - \epsilon < h^i(x'').$$

Clearly, PSG is a distance-based discrepancy metric that motivated from a purely geometric viewpoint. In practice, the comparator set $\mathcal{K}^*$ is often set to be the Pareto set $\mathcal{P}_{\mathcal{K}}(H)$ [30]. Then for any $x \in \mathcal{K}$, its PSG is always non-negative and equals to zero if and only if $x \in \mathcal{P}_{\mathcal{K}}(H)$.

**Multiple Gradient Descent Algorithm (MGDA)** is an offline first-order algorithm for MOO [9, 7]. At each iteration $l \in \{1, \ldots, L\}$ ($L$ is the number of iterations), it first computes the gradient $\nabla h^i(x_l)$ for each objective $i \in \{1, \ldots, m\}$, then derive the composite gradient $g_l^{comp} = \sum_{i=1}^m \lambda_l^i \nabla h^i(x_l)$ as the convex combination of these multiple gradients; it applies $g_l^{comp}$ to execute the gradient descent step to update the decision, i.e., $x_{l+1} = x_l - \eta g_l^{comp}$, where $\eta$ is the step size. The core part of MGDA is the module that determines the composite weights $\lambda_l = (\lambda_l^1, \ldots, \lambda_l^m)$, which is given as

$$\lambda_l = \arg \min_{\lambda_l \in \mathcal{S}_m} \| \sum_{i=1}^m \lambda_l^i \nabla h^i(x_l) \|_2^2,$$

where $\mathcal{S}_m = \{\lambda \in \mathbb{R}^m \mid \sum_{i=1}^m \lambda^i = 1, \lambda^i \geq 0, i \in \{1, \ldots, m\}\}$ denotes the probabilistic simplex in $\mathbb{R}^m$. This is a min-norm solver which finds the weights in the simplex that yields the minimum $L_2$ norm of the composite gradient. Thus MGDA is also called the *min-norm* method. Existing works [7, 29] have shown that MGDA is guaranteed to decrease all the objectives simultaneously until it reaches a Pareto optimal decision (under the convex setting where all $h^i$ are convex functions).

# 3 Multi-Objective Online Convex Optimization

In this section, we formally formulate the framework of multi-objective optimization in the online setting, termed Multi-Objective Online Convex Optimization (MO-OCO).

**Framework overview.** We tailor the famous online convex optimization (OCO) framework to the multi-objective setting, which can be viewed as a repeated game between an online learner and the adversarial environment. At each round $t \in \{1, \ldots, T\}$, the learner generates a decision $x_t$ from a given convex compact decision set $\mathcal{X} \subset \mathbb{R}^n$. Then the adversary replies the decision with a vector loss function $F_t(x) : \mathcal{X} \to \mathbb{R}^m$, where its $i$-th component $f_t^i(x) : \mathcal{X} \to \mathbb{R}$ belongs to the $i$-th objective, and the learner suffers the loss $F_t(x_t) \in \mathbb{R}^m$. The goal of the learner is to generate a sequence of decisions $\{x_t\}_{t=1}^T$ so that the cumulative loss $\sum_{t=1}^T F_t(x_t)$ can be optimized.

Recall that, in the single-objective setting, the performance metric $R(T) = \sum_{t=1}^T (f_t(x_t) - f_t(z_t))$, i.e., the regret, compares the actual decisions $x_t$ with some comparator $z_t \in \mathcal{X}$ at each round $t$. For the static regret, all $z_t$ are identically set as the fixed optimal decision $x^*$ w.r.t. all losses in hindsight, i.e., $z_t \equiv x^* \in \arg \min_{x \in \mathcal{X}} \sum_{t=1}^T f_t(x)$. For the dynamic regret, each $z_t$ is selected as the optimal decision $x_t^*$ w.r.t. the instantaneous loss $f_t$ at that round, i.e., $z_t = x_t^* \in \arg \min_{x \in \mathcal{X}} f_t(x)$.

In analogy, we can define the multi-objective regret as $R(T) = \sum_{t=1}^T \Delta_t$, where each $\Delta_t$ compares the actual decisions $x_t$ with some comparator $z_t \in \mathcal{X}$. However, in general, no single decision can optimize all the objectives at the same time. Hence, it is natural to compare $x_t$ with a group of Pareto optimal decisions, which constitute a comparator set $\mathcal{C}_t \subset \mathcal{X}$. To measure the discrepancy between $x_t$ and $\mathcal{C}_t$, we further introduce the Pareto suboptimality gap (PSG) [30] $\Delta(x_t; \mathcal{C}_t, F_t)$. Then the multi-objective regret can be defined as $R(T) = \sum_{t=1}^T \Delta(x_t; \mathcal{C}_t, F_t)$. Now we can formulate the static or the dynamic variant by specifying the comparator set $\mathcal{C}_t$ at each round. Specifically, by setting all $\mathcal{C}_t$ to be the Pareto set $\mathcal{X}^*$ of the cumulative loss $\sum_{t=1}^T F_t$, we formulate the **multi-objective static regret** $R_{\text{MOS}}(T) = \sum_{t=1}^T \Delta(x_t; \mathcal{X}^*, F_t)$. By setting each $\mathcal{C}_t$ to be the Pareto set $\mathcal{X}_t^*$ of the instantaneous loss $F_t$, we formulate the **multi-objective dynamic regret** $R_{\text{MOD}}(T) = \sum_{t=1}^T \Delta(x_t; \mathcal{X}_t^*, F_t)$.

---

[1]Our definition of PSG is a bit different from that in [30]. In Appendix B we show that they are equivalent.

Recall that PSG is a zero-order metric motivated in a purely geometric sense, namely, its calculation needs to solve a constrained optimization problem with an unknown boundary $f_t^i(x''), \forall x'' \in \mathcal{C}_t$. Hence, it is not straightforward to design a first-order algorithm to optimize PSG, not to mention the regret analysis. To motivate algorithm design and analysis, we investigate the two variants in more detail. We begin with the dynamic variant, since we find that it has an equivalent form, which is intuitive and has a strong implication on the design of effective online multiple gradient algorithms.

**An equivalent form of the dynamic regret.** Surprisingly, the multi-objective dynamic regret $R_{\text{MOD}}$ can be transformed into an unconstrained max-min form. The derivation utilizes Pareto optimality of $\mathcal{X}_t^*$ and is highly non-trivial, which is deferred to the appendix due to the space limit.

**Proposition 3.1.** *The multi-objective dynamic regret has an equivalent form, i.e.,*

$$R_{\text{MOD}}(T) = \sup_{\substack{x_t^* \in \mathcal{X}_t^*, \\ 1 \le t \le T}} \inf_{\substack{\lambda_t^* \in \mathcal{S}_m, \\ 1 \le t \le T}} \sum_{t=1}^{T} \lambda_t^{*\top} (F_t(x_t) - F_t(x_t^*)).$$

*Remark.* (i) The above form can be understood as a variant of the standard dynamic regret regarding $\{\lambda_t^{*\top} F_t\}_{t=1}^T$, whereas $\lambda_t^*$ are unknown to the learner. This provides an intuition that we can generate weights $\lambda_t \in \mathcal{S}_m$ at each round and optimize $\{\lambda_t F_t\}_{t=1}^T$ via single-objective techniques. For first-order algorithms, it is equivalent to selecting a convex combination of individual gradients and then applying the composite gradient to model update. Undoubtedly, how to generate the weights $\lambda_t$ needs some careful designs, which will be explicated later in the algorithm section.
(ii) When $m = 1$, we have $\mathcal{S}_m = \{1\}$ and $\mathcal{X}_t^* = \arg\min_{x \in \mathcal{X}} F_t(x)$. Hence $R_{\text{MOD}}(T) = \sum_{t=1}^{T}(F_t(x_t) - \min_{x \in \mathcal{X}} F_t(x))$, which is exactly the single-objective dynamic regret $R_D(T)$.

**An alternative form of the static regret.** Unfortunately, for $R_{\text{MOS}}$, the above equivalence form does not exist. Here is the reason. In $R_{\text{MOS}}$, the comparator set $\mathcal{X}^*$ is the Pareto set of the cumulative loss $\sum_{t=1}^{T} F_t$ rather than the instantaneous loss $F_t$. Hence, at some specific round $t$, the decision $x_t$ may Pareto dominate all points in $\mathcal{X}^*$ w.r.t. the instantaneous $F_t$, and we would expect the metric $\Delta_t$ to be negative. However, PSG (or other commonly used metrics such as Hypervolume) always yields non-negative values, so the induced $R_{\text{MOS}}$ is not aligned with $R_S$. For example, when $m = 1$, we have $R_{\text{MOS}}(T) = \sup_{x^* \in \mathcal{X}^*} \sum_{t=1}^{T} \max\{F_t(x_t) - F_t(x^*), 0\}$, which can be much looser than the static regret $R_S(T) = \sup_{x^* \in \mathcal{X}^*} \sum_{t=1}^{T}(F_t(x_t) - F_t(x^*))$. Hence the analysis of $R_{\text{MOS}}$ is intrinsically complex if we use existing discrepancy metrics that always yield non-negative values.

Enlightened by Proposition 3.1, we can formulate the static regret in a different way, i.e., by modifying the equivalent form of dynamic regret. Recall that in Proposition 3.1, at each round $t$, the comparator $x_t^*$ is selected from the Pareto set $\mathcal{X}_t^*$ of the instantaneous loss $F_t$, and the weights $\lambda_t^*$ are generated from $\mathcal{S}_m$. To formulate the static variant, we can use a fixed comparator $x^*$ from the Pareto set $\mathcal{X}^*$ of the cumulative loss $\sum_t F_t$ and fixed weights $\lambda^* \in \mathcal{S}_m$ at all rounds. Now the static variant takes

$$R_{\text{MOS}}(T) := \sup_{x^* \in \mathcal{X}^*} \inf_{\lambda^* \in \mathcal{S}_m} \lambda^{*\top}(\sum_{t=1}^{T} F_t(x_t) - \sum_{t=1}^{T} F_t(x^*)).$$

*Remark.* (i) $R_{\text{MOS}}(T)$ has a clear physical meaning that optimizing it will impose the cumulative loss $\sum_{t=1}^{T} F_t(x_t)$ to reach the Pareto front $\mathcal{P}^*$. See more details in Appendix C.
(ii) When $m = 1$, $\mathcal{S}_m = \{1\}$ and $\mathcal{X}^*$ reduces to $\arg\min_{x \in \mathcal{X}} \sum_{t=1}^{T} F_t(x)$. Therein $R_{\text{MOS}}(T) = \sum_{t=1}^{T} F_t(x_t) - \min_{x^* \in \mathcal{X}^*} \sum_{t=1}^{T} F_t(x^*)$, which reduces to the single-objective static regret $R_S(T)$.

# 4 Online Mirror Multiple Descent

In this section, we present the Online Mirror Multiple Descent (OMMD) algorithm, the protocol of which is given in Algorithm 1. At each round $t$, the learner first computes the gradient of the loss regarding each objective, then determines the composite weights of all these gradients, and finally applies the composite gradient to the online mirror descent step.

## 4.1 Vanilla Min-Norm May Incur Linear Regrets

The core module of OMMD is the composition of multiple gradients. For simplicity, we represent the gradients at round $t$ in a matrix form $\nabla F_t(x_t) = [\nabla f_t^1(x_t), \ldots, \nabla f_t^m(x_t)] \in \mathbb{R}^{n \times m}$. Then the

**Algorithm 1** Doubly Regularized Online Mirror Multiple Descent **(DR-OMMD)**

---

1: **Input:** Convex set $\mathcal{X}$, time horizon $T$, regularization parameter $\alpha_t$, learning rate $\eta_t$, regularization function $R$, user preference $\lambda_0$.
2: **Initialize:** $x_1 \in \mathcal{X}$.
3: **for** $t = 1, \ldots, T$ **do**
4:    Predict $x_t$ and receive a loss function $F_t : \mathcal{X} \to \mathbb{R}^m$.
5:    Compute the multiple gradients $\nabla F_t(x_t) = [\nabla f_t^1(x_t), \ldots, \nabla f_t^m(x_t)] \in \mathbb{R}^{n \times m}$.
6:    Determine the weights for the gradient composition via **min-regularized-norm**
$$\lambda_t = \arg \min_{\lambda \in \mathcal{S}_m} \|\nabla F_t(x_t)\lambda\|_2^2 + \alpha \|\lambda - \lambda_0\|_1.$$
7:    Compute the composite gradient $g_t = \nabla F_t(x_t)\lambda_t$.
8:    Perform online mirror descent using $g_t$
$$x_{t+1} = \arg \min_{x \in \mathcal{X}} \eta \langle g_t, x \rangle + B_R(x, x_t).$$
9: **end for**

---

composite gradient is given as $g_t = \nabla F_t(x_t)\lambda_t$, where $\lambda_t$ is the composite weights. As illustrated in Preliminary, the min-norm method in MGDA [7, 29] is a classic method to determine the composite weights in the offline setting, which results in a common descent direction that can descend all the losses simultaneously. Thus, it is tempting to consider applying it to the online setting.

However, directly applying the min-norm method to the online setting is not workable, which may even incur linear regrets of the resulting algorithms. The rationale is as follows. In the vanilla min-norm method, the composite weights $\lambda_t$ are determined solely by the gradients $\nabla F_t(x_t)$ at the current round $t$, hence they are very sensitive to the instantaneous loss $F_t$. In the online setting, the losses at each round can be adversarially chosen, and thus the corresponding gradients can be adversarial. These adversarial gradients may result in undesired composite weights, which may further produce a composite gradient that even deteriorates the next prediction. In the following, we provide a problem instance in which min-norm incurs a linear regret. We extend OMD to the multi-objective setting, where the composite weights are directly yielded by min-norm [11].

**Problem instance.** We consider a two-objective problem. The decision domain is $\mathcal{X} = \{(u, v) \mid u + v \leq \frac{1}{2}, v - u \leq \frac{1}{2}, v \geq 0\}$ and the loss function at each round is

$$F_t(x) = \begin{cases} (\|x - a\|^2, \|x - b\|^2), & t = 2k - 1, \quad k = 1, 2, \ldots; \\ (\|x - b\|^2, \|x - c\|^2), & t = 2k, \quad k = 1, 2, \ldots, \end{cases}$$

where $a = (-2, -1), b = (0, 1), c = (2, -1)$. For simplicity, we first analyze the case where the total time horizon $T$ is an even number. Then we can compute the Pareto set of the cumulative loss $\sum_{t=1}^T F_t$, i.e., $\mathcal{X}^* = \{(u, 0) \mid -\frac{1}{2} \leq u \leq \frac{1}{2}\}$, which locates at the $x$-axis. For conciseness of analysis, we instantiate OMD with L2-regularization, which results in the simple OGD algorithm [24]. We start at an arbitrary point $x_1 = (u_1, v_1) \in \mathcal{X}$ satisfying $v_1 > 0$. At each round $t$, suppose the decision $x_t = (u_t, v_t) \in \mathcal{X}$, then the gradients of each objective w.r.t. $x_t$ can be calculated as

$$g_t^1 = \begin{cases} (2u_t + 4, & 2v_t + 2), & t = 2k - 1; \\ (2u_t, & 2v_t - 2), & t = 2k. \end{cases} \qquad g_t^2 = \begin{cases} (2u_t, & 2v_t - 2), & t = 2k - 1; \\ (2u_t - 4, & 2v_t + 2), & t = 2k. \end{cases}$$

Since $0 \leq v_t \leq \frac{1}{2}$, we observe that the second entry of either gradient alternates between positive and negative. By using min-norm, the composite weights $\lambda_t$ can be computed as

$$\lambda_t = \begin{cases} ((1 - u_t - v_t)/4, & (3 + u_t + v_t)/4), & t = 2k - 1; \\ ((3 - u_t + v_t)/4, & (1 + u_t - v_t)/4), & t = 2k. \end{cases}$$

We observe that both entries of composite weights alternative between above $\frac{1}{2}$ and below $\frac{1}{2}$, and $\|\lambda_{t+1} - \lambda_t\|_1 \geq 1$. Recall that $\|\lambda_t\|_1 = 1$, hence the composite weights at two consecutive rounds change radically. The resulting composite gradient takes

$$g_t^{comp} = \begin{cases} (u_t - v_t + 1, & -u_t + v_t - 1), & t = 2k - 1; \\ (-u_t - v_t - 1, & -u_t - v_t - 1), & t = 2k. \end{cases}$$

The fluctuating composite weights mix with the positive and negative second entries of gradients, making the second entry of $g_t^{comp}$ always negative, i.e., $-u_t + v_t - 1 < 0$ and $-u_t - v_t - 1 < 0$. Hence $g_t^{comp}$ actually drives $x_t$ away from the Pareto set $\mathcal{X}^*$ that coincides with the $x$-axis. This essentially reversely optimizes the loss, hence increases the regret. In fact, we can prove that it even incurs a linear regret[2]. Due to the lack of space, we leave the proof of linear regret when $T$ is an odd number in the appendix. The above results of the problem instance are summarized as follows.

**Proposition 4.1.** *For OMD equipped with vanilla min-norm, there exists a multi-objective online convex optimization problem, in which the resulting algorithm incurs a linear regret.*

***Remark.*** Stability is a basic requirement to guarantee meaningful regrets in online learning [25]. In the single-objective setting, directly regularizing the iterate $x_t$ (e.g., OMD) is already enough. However, as shown in the above analysis, only regularizing $x_t$ is not enough to attain sublinear regrets in the multi-objective setting, since there is another source of instability, i.e., the composite weights, that affects the direction of the composite gradient. Therefore, in multi-objective online learning, besides regularizing the iterates, we also need to explicitly regularize the composite weights.

## 4.2 Doubly Regularized Online Mirror Multiple Descent

Enlightened by the design of regularization in FTRL [25], we consider the regularizer $r(\lambda, \lambda_0)$, where $\lambda_0$ is the pre-defined composite weight that may reflect the user preference. This results in a new solver called *min-regularized-norm*, i.e.,

$$\lambda_t = \underset{\lambda \in \mathcal{S}_m}{\arg\min} \|\nabla F_t(x_t)\lambda\|_2^2 + \alpha r(\lambda, \lambda_0),$$

where $\alpha$ is the strength of regularization. Equipping OMD with the new solver, we derive the proposed online algorithm. Note that beyond the regularization on the iterate $x_t$ that is intrinsic in online learning, there is another regularization on the composite weights $\lambda_t$ in min-regularized norm. Both regularizations are fundamental and they together ensure the stability in the multi-objective online setting. Hence we call the algorithm Doubly Regularized OMMD (DR-OMMD).

In principle, $r$ can take various forms such as $L_1$-norm, $L_2$-norm and KL divergence etc. Here we adopt $L_1$-norm since it aligns well with the simplex constraint of $\lambda$. Min-regularized-norm can be computed very efficiently, since it has a closed-form solution when $m = 2$. Specifically, suppose the gradients at round $t$ are $g_t^1$ and $g_t^2$. Set $\gamma_L = (g_2^\top(g_2 - g_1) - \alpha)/\|g_2 - g_1\|^2$ and $\gamma_R = (g_2^\top(g_2 - g_1) + \alpha)/\|g_2 - g_1\|^2$. Given any $\lambda_0 = (\gamma_0, 1 - \gamma_0) \in \mathcal{S}_2$, we can compute the composite weights $\lambda_t$ as $(\gamma_t, 1 - \gamma_t)$ where

$$\gamma_t = \max\{\min\{\gamma_t'', 1\}, 0\}, \quad \text{where } \gamma_t'' = \max\{\min\{\gamma_0, \gamma_R\}, \gamma_L\}.$$

In addition, when $m > 2$, since the feasible region $\mathcal{S}_m$ is a simplex, we can introduce a Frank-Wolfe solver [14] to compute the composite weights. See the protocol and more details in Appendix D.

Compared to vanilla min-norm, the composite weights in min-regularized-norm are not fully determined by the adversarial gradients. The resulting relative stability of composite weights make the composite gradients more robust to the adversarial environment. In the following, we give a general analysis and prove that DR-OMMD indeed guarantees sublinear regrets.

## 4.3 Analysis

We now analyze the static regret and the dynamic regret of DR-OMMD. Our analysis is based on the following commonly used assumptions [13, 11].

**Assumption 4.2 (Bregman divergence).** The regularization function $R$ is 1-strongly convex. In addition, the Bregman divergence is $\gamma$-Lipschitz continuous, i.e., $B_R(x, z) - B_R(y, z) \leq \gamma \|x - y\|, \forall x, y, z \in \text{dom}R$, where $\text{dom}R$ is the domain of $R$ and satisfies $\mathcal{X} \subset \text{dom}R \subset \mathbb{R}^n$.

**Assumption 4.3 (Lipschitz continuity).** For each $i \in \{1, \ldots, m\}$, there exists some positive and finite $G$ such that, the $i$-th loss $f_t^i$ at each round $t \in \{1, \ldots, T\}$ is $G$-Lipschitz continuous w.r.t. $\|\cdot\|$, i.e., $|f_t^i(x) - f_t^i(x')| \leq G\|x - x'\|$. Note that in the convex setting, this assumption leads to bounded gradients, i.e., $\|\nabla f_t^i(x)\|_* \leq G$ for any $t \in \{1, \ldots, T\}, i \in \{1, \ldots, m\}, x \in \mathcal{X}$.

---

[2]More concisely, here the regret is the multi-objective static regret.

We first provide the static regret bound. The proof is left to the appendix due to the lack of space.

**Theorem 4.4.** *Suppose the diameter of $\mathcal{X}$ is bounded by $D$. Assume $F_t$ is bounded, i.e., $|f_t^i(x)| \leq F, \forall x \in \mathcal{X}, t \in \{1, \ldots, T\}, i \in \{1, \ldots, m\}$. For any $\lambda_0 \in \mathcal{S}_m$, DR-OMMD attains*

$$R_{\text{MOS}}(T) \leq \frac{1}{\eta} B_R(x^*, x_1) + \frac{\eta}{2} \sum\nolimits_{t=1}^{T} (\|\nabla F_t(x_t)\lambda_t\|_2^2 + \frac{4F}{\eta} \|\lambda_t - \lambda_0\|_1).$$

*Remark.* (i) Linearization with weights $\lambda_0 \in \mathcal{S}_m$ can be viewed as single-objective optimization on scalar loss $\lambda_0^\top F_t$, whose gradient is $g_t = \nabla F_t(x_t)\lambda_0$. Hence we can directly borrow the tight bound of OMD (Theorem 6.8 in [27]) and derive a bound $\frac{1}{\eta} B_R(x^*, x_1) + \sum_{t=1}^{T} \frac{\eta_t}{2} \|\nabla F_t(x_t)\lambda_0\|_2^2$ for linearization. In comparison, if we set $\alpha = 4F/\eta$ in DR-OMMD, then from the formulation of $\lambda_t$, the bound becomes $\frac{1}{\eta} B_R(x^*, x_1) + \frac{\eta}{2} \sum_{t=1}^{T} \min_{\lambda \in \mathcal{S}_m} \{\|\nabla F_t(x_t)\lambda\|^2 + \alpha \|\lambda - \lambda_0\|_1\}$, which is smaller than that of linearization. Note that the lower regret of DR-OMMD compared to linearization is also empirically verified in our experiments (see Figure 1).

(ii) When $\eta = \frac{\sqrt{2\gamma D}}{G\sqrt{T}}, \alpha = \frac{4F}{\eta}$, the bound is in the order of $O(\sqrt{T})$. It matches the optimal static single-objective regret bound w.r.t. $T$ [11] (see more details in Appendix E).

Then we turn to the dynamic regret. Our analysis relies on an additional assumption [2, 32, 5].

**Assumption 4.5** (**Temporal variability**). *For each $i \in \{1, \ldots, m\}$, there exists some positive and finite $V_T$ such that $\sum_{t=1}^{T-1} \sup_{x \in \mathcal{X}} |f_t^i(x) - f_{t+1}^i(x)| \leq V_T$.*

**Theorem 4.6.** *Assume the step size satisfies $\frac{4V_T}{G^2 T} \leq \eta \leq \frac{4V_T}{G^2}$. Then under all the above assumptions, for any preference $\lambda_0 \in \mathcal{S}_m$, OMMD with min-regularized-norm attains*

$$R_{\text{MOD}}(T) \leq \frac{\eta G^2 T}{2} + \frac{4\gamma D V_T}{\eta^2 G^2} + \frac{\eta}{2} \sum\nolimits_{t=1}^{T} (\|\nabla F_t(x_t)\lambda_t\|_2^2 + \frac{8FG^2 T}{V_T} \|\lambda_t - \lambda_0\|_1).$$

*Remark.* When $\eta = \frac{2}{G} (\frac{\gamma D V_T}{GT})^{1/3}, \alpha = \frac{8FG^2 T}{V_T}$, the bound is in the order of $O(T^{2/3} V_T^{1/3})$, matching the best attainable single-objective dynamic regret bound [2, 35] (see more details in Appendix E).

# 5 Experiments

In this section, we conduct extensive experiments to evaluate the effectiveness of DR-OMMD. We consider two baselines: (i) *linearization* performs single-objective online learning on the linearized loss $\lambda_0^\top F_t$ at each round $t$, where the weights $\lambda_0 \in \mathcal{S}_m$ are given beforehand; note that it is equivalent to computing composite gradients with fixed weights $\lambda_t \equiv \lambda_0$. (ii) *min-norm* equips OMD with vanilla min-norm [7] for gradient composition.

## 5.1 Simulation Experiments: Tracking the Pareto Front

As summarized in Figure 1 (a), the goal is to track two points $\xi_t^1, \xi_t^2$ cycling along a circle $\mathcal{C} = \{\xi \in \mathbb{R}^2 \mid \|\xi\|_2 = 1\}$. For each $i \in \{1, 2\}$, $\xi_t^i = (\cos \theta_t^i, \sin \theta_t^i)$ is determined by some angle $\theta_t^i$. We set a positive integer $P^i$ as the rotating period of $\xi_t^i$, which is unknown to the learner. The two points are initialized by $\theta_1^1 = 0$ and $\theta_1^2 = \pi/2$ and move as follows: at each round $t$, for each $i \in \{1, 2\}$, the adversary independently samples an angle $\delta_t^i$ from a Gaussian distribution $\mathcal{N}(2\pi/P^i, 1/\sqrt{P^i})$, then moves the $i$-th point to $\xi_{t+1}^i = (\cos \theta_{t+1}^i, \sin \theta_{t+1}^i)$ where $\theta_{t+1}^i = \theta_t^i - \delta_t^i$. Note that $\mathbb{E}\theta_{t+1}^i = \theta_1^i + 2\pi t/P^i$, hence in average $\xi_t^i$ rotates clockwise with a period of $P^i$. At each round $t$, the learner generates a decision $x_t$ from a $L2$-norm ball $\mathcal{X} = \{x \in \mathbb{R}^2 \mid \|x\|_2 \leq 2\}$. Then it acquires $\xi_t^1, \xi_t^2$ and suffer the losses $f_t^i(x_t) = \|x_t - \xi_t^i\|_2^2/2, i \in \{1, 2\}$. In this problem, the Pareto set of $F_t = (f_t^1, f_t^2)$ is exactly the line segment between $\xi_t^1$ and $\xi_t^2$, i.e., $\mathcal{X}_t^* = \{\lambda \xi_t^1 + (1 - \lambda)\xi_t^2 \mid \lambda \in [0, 1]\}$. At each round $t$, PSG measures the squared distance between $x_t$ and $\mathcal{X}_t^*$.

We run $T = 10,000$ rounds. To simulate the pattern drift, we set $P^1 = 10, P^2 = 20$ at the first $T_1 = 3,000$ rounds, and $P^1 = 20, P^2 = 10$ at the last $T_2 = 7,000$ rounds. For linearization, the weights $\lambda_0 = (\lambda_0^1, 1 - \lambda_0^1)$ are decided via a grid search $\lambda_0^1 \in \{0, 0.1, ..., 1\}$; we consider three variants: *lin-1* uses the optimal $\lambda_0$ for the first $T_1$ rounds, *lin-2* uses the optimal $\lambda_0$ for the last $T_2$ rounds, and *lin-opt* uses the optimal $\lambda_0$ for all $T$ rounds. For DR-OMMD, for fairness of

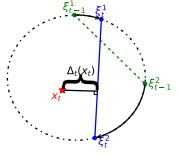

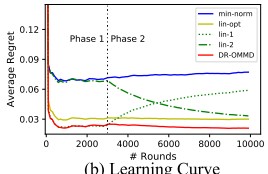

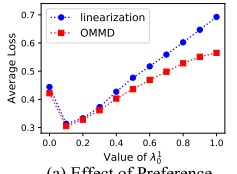

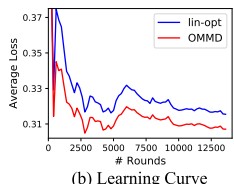

| (a) Simulation Setup | (b) Learning Curve | (a) Effect of Preference | (b) Learning Curve |

Figure 1: Simulation setup and results. (a) The targets $\xi_t^1, \xi_t^2$ cycle along the circle. The Pareto set at each round is the line segment $[\xi_t^1, \xi_t^2]$; PSG measures the distance from $x_t$ to $[\xi_t^1, \xi_t^2]$. (b) Performance of DR-OMMD and baselines.

Figure 2: Results to verify the effectiveness of adaptive regularization on *protein*. (a) Performance of DR-OMMD and linearization under varying $\lambda_0 = (\lambda_0^1, 1 - \lambda_0^1)$. (b) Performance using the optimal weights $\lambda_0 = (0.1, 0.9)$.

comparison we use the same $\lambda_0$ of *lin-opt*. The learning rates $\eta$ in all algorithms and the parameter $\alpha$ in DR-OMMD follow the corresponding theories (e.g., Theorem 4.6). In this experiment, since the loss functions are manually designed, the value of $V_T$ can be directly calculated. Note that in some scenarios where $V_T$ is unknown, we can conduct a grid search and utilize a meta-algorithm to handle the unknown $V_T$ [37, 1], similar to the single-objective setting. From the results in Figure 1 (b), we find that DR-OMMD achieves the lowest PSG, showing its ability to track the Pareto front; meanwhile, *min-norm* appears very unstable in the online setting, even worse than linearization.

## 5.2 Convex Experiments: Adaptive Regularization via Multi-Objective Optimization

In many real-world online scenarios, regularization is often adopted to avoid overfitting. A standard way is to add a term $r(x)$ to the loss $f_t(x)$ at each round and optimize the regularized loss $f_t(x) + \sigma r(x)$ [24], where $\sigma$ is treated as a hyperparameter that needs to be fixed beforehand. The formalism of multi-objective online learning provides a novel way to realize regularization. Since $r(x)$ measures the complexity of $x$, it can be regarded as the second objective alongside the primary goal $f_t(x)$. We can construct a vector loss $F_t(x) = (f_t(x), r(x))$ at each round and thereby cast regularized online learning into a bi-objective online optimization problem. Compared to fixed regularization, the new approach effectively chooses the regularization strength $\sigma_t = \lambda_t^2/\lambda_t^1$ in an adaptive way.

We use two large-scale online benchmark datasets. (i) *protein* is a bioinformatics dataset for protein type classification [31], which has 17 thousand instances with 357 features. (ii) *covtype* is a biological dataset collected from a non-stationary environment for forest cover type prediction [3], which has 50 thousand instances with 54 features. For both tasks, we set the logistic loss of classification as the first objective, and the squared $L2$-norm of model parameters as the second objective. Since the ultimate goal of regularization is to enhance predictive performance, we adopt the average loss as the performance metric, namely $\sum_{t \le T} l_t(x_t)/T$, where $l_t(x_t)$ is the classification loss at round $t$.

We adopt a $L2$-norm ball centered at the origin with diameter $K = 100$ as the decision set. The learning rates are decided by a grid search over $\{0.1, 0.2, \ldots, 3.0\}$. For DR-OMMD, the parameter $\alpha$ is simply set as 0.1. For fixed regularization, the strength $\sigma = (1 - \lambda_0^1)/\lambda_0^1$ is determined by the some preference $\lambda_0^1 \in [0, 1]$, which is essentially *linearization* with weights $\lambda_0 = (\lambda_0^1, 1 - \lambda_0^1)$. We run both algorithms with varying initial weights $\lambda_0^1 \in \{0, 0.1, ..., 1\}$. In Figure 2, we plot (a) their final performance w.r.t. the choice of $\lambda_0$ and (b) their learning curves with desirable $\lambda_0$ (e.g., $(0.1, 0.9)$ on *protein*). Other results are deferred to the appendix due to the lack of space. The results show that DR-OMMD consistently outperforms fixed regularization.

## 6 Conclusions

In this paper, we give a systematic study of multi-objective optimization in the online setting. We first formulate the framework of Multi-Objective Online Convex Optimization. Then we devise the Doubly Regularized Online Mirror Multiple Descent algorithm, which has a special design for gradient composition in online learning, namely min-regularized-norm. We provide non-trivial regret bounds for DR-OMMD and conduct extensive experiments to demonstrate its effectiveness.

**Limitations.** As the first step of studying multiple gradient algorithm in online learning, we conduct our analysis in the convex setting. Although it does not affect the usage in the non-convex setting (see empirical validation in Appendix F), we can give a formal non-convex analysis in the future.

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
