# OpenReview forum: "Multi-Objective Online Learning"
_NeurIPS.cc/2022/Conference — NeurIPS 2022 Submitted_

### Official Review · Reviewer_aTnA · 2022-07-10

**Rating:** 3
**Confidence:** 3
**Soundness:** 2 fair
**Presentation:** 3 good
**Contribution:** 2 fair

**Summary:**

This paper studies the online multi-objective learning problem in the online convex optimization (OCO) framework. The main contribution of the paper is to show that the regret minimization problem over the Pareto suboptimal gap can be converted to a single-objective optimization problem. Then, the online mirror descent method can be applied to achieve the static and dynamic regret bound for the multi-objective learning problem.


**Questions:**

- What is the gap between the alternative form of static regret (Line 210) and the static regret defined by PSG (Line 177)?  Can the gap between the two regrets be bounded?
- it seems the code in the supplementary material only contains the implementation of the min-norm solver rather than the whole code.
- the optimal step size in Theorem 4.6. relies on the temporal variability $V_T$, which is unknown in advance. In the single-objective setting, some previous works try to solve this issue by grid search with a meta-algorithm [4,5,6]. How does the algorithm in this paper handle the non-stationarity to obtain the dynamic regret?
- in Line 42-43, the authors have claimed that “the theoretical investigation of multi-objective online learning will lay a solid foundation for the design of new optimizers for multi-task deep neural networks.” However, it seems to me that the theoretical results in this paper do not have a strong connection with multi-task DNN ( in OCO framework, the function should always be convex). It would be better if the author could elaborate more on this issue.


[4] P. Zhao, Y.-J. Zhang, L. Zhang, Z.-H. Zhou. Dynamic regret of convex and smooth functions. In: Advances in Neural Information Processing Systems 33 (NeurIPS), 2020.

[5] D. Baby and Y.-X. Wang. Optimal dynamic regret in exp-concave online learning. In: Proceedings of the 34th Conference on Learning Theory (COLT), 2021.

[6] D. Baby and Y.-X. Wang. Optimal Dynamic Regret in Proper Online Learning with Strongly Convex Losses and Beyond. In: Proceedings of the International Conference on Artificial Intelligence and Statistics (AISTATS), 2022.


**Limitations:**

This paper has discussed its potential limitations in Line 370-372. But I think there are also some other limitations as I have listed in the weakness and question part.

**Strengths And Weaknesses:**

### Strength:
- Novelty: although the definition of Pareto suboptimal gap (PSG) has appeared in the bandits setting, the paper proposes an interesting reduction in the full information setting that the regret defined by PSG can be converted to a single-object optimization problem.
- Clarity: this paper is well-structured and clearly written in most parts, which makes it easy to follow.

### Weakness:
- the title of this paper is somewhat overclaimed to me. The focus of this paper is on the multi-objective optimization in OCO (full information), and a gradient-based algorithm is proposed, which is not that general enough to cover the entire multi-objective online learning. For example, as referred by the paper, there are many papers on multi-object optimization in the context of bandit. Besides, multi-objective learning does not only exactly mean multi-objective optimization. For instance, [1] considers the optimization of two objectives in MAB (best arm identification and regret minimization); other results can be found in the references therein. Even in the full information, there is work studying how to handle the Pareto-frontier in the absolute loss game [2]. So, I advise the author to place the paper in the right place of the contemporary literature to avoid misleading.

- Although the reduction proposed in this paper is interesting, I still have some reservations about the main theorems and their proofs.
	- on Theorem 4.6: by the optimal step size, the method achieves an $O(T^{2/3}V_T^{1/3})$ dynamic regret bound. But in the full information online learning (i.e. there is no noise in the feedback), a simple algorithm by just playing $x_t^* = \\arg \\min_{x\\in\\mathcal{X}} f_{t-1}(x)$ can achieve $O(V_T)$ dynamic regret [3]. I am not sure whether the $O(T^{2/3}V_T^{1/3})$ bound is tight in the full information case. It would be nice if the author could provide a lower bound.
	- another concern is the technical contribution of this paper. It seems that the proofs of Theorem 4.4 and Theorem 4.6 follow the standard analysis of the OMD algorithm. I think it would be nice for the author to highlight their technique contributions.

- about the experiments: as shown by the authors in line 337, the learning rate $\eta$ and parameter $\alpha$ are set according to the theories. However, it seems that the setting of the step size $\eta$ requires the knowledge of $V_T$, which is unknown in advance in the practice. So, it is unclear to me how the authors choose the parameter in the experiments ( Maybe one can calculate the $V_T$ in advance of the online learning process, but this could make a somewhat artificial experiment setup.).

- about the reproducibility: it seems that only the supplementary material only provides the code on the min-norm solver, which is not the method proposed by this paper.

[1] R. Degenne, T. Nedelec, C. Calauzenes and V. Perchet. Bridging the gap between regret minimization and best arm identification. In: AISTATS 2019.

[2] Koolen W M. The pareto regret frontier. In: NeurIPS 2013.

[3] Yang, T., Zhang, L., Jin, R., and Yi, J. (2016, June). Tracking slowly moving clairvoyant: Optimal dynamic regret of online learning with true and noisy gradient. In: ICML 2016.

---

> ### Author Response · Authors · 2022-08-01
> **Author Response (Part 3)**
>
> **Q5. "only provide the code on the min-norm solver"**
>
> We have provided the main code of min-regularized-norm. This code is under the name of *find\_min\_norm\_element\_l1*. To use this solver, you can simply plug it into any basic optimizer and data-processing modules. In the revision, we have added some remarks to our code to make it easier to understand.
>
> **Q6. The gap between the two static regret forms. Why use the alternative form for static regret?**
>
> As we have explained in Section 3 (lines 197-205), the vanilla PSG-based static regret is typically much larger than the alternative form. The gap between them can even be **LINEAR** in some cases, which indicates that the vanilla form is unsuitable as a regret metric. To better explicate these points, let us consider the case of $m=1$. In this simple case, PSG-based regret reduces to $R^{psg}={\rm sup}\_{x^\*\in\mathcal X^\*}\sum_t\max\\{f^1_t(x_t)-f^1_t(x^\*), 0\\}$, and the alternative form takes $R^{alt}=\sup_{x^\*\in\mathcal X^\*}\sum_t(f^1_t(x_t)-f^1_t(x^\*))$. For $R^{alt}$, it directly recovers the single-objective static regret $\sum_t f^1_t(x_t)-\inf_{x^\*\in\mathcal X^\*}\sum_t f^1_t(x^\*)$. Notably, $R^{alt}$ equals to zero with optimal decisions $x_t=x^\*\in\mathcal X^\*$. For $R^{psg}$, however, due to the irregularity of the maximum operation, it may even be **LINEAR** when the decisions $x_t$ are already optimal, which is problematic as a regret metric. In the following, we provide a detailed example.
>
> Set $\mathcal X=[-2,2]$ and let the loss function $f^1_t(x)$ alternates between $x$ and $-x$. Assume the time horizon $T$ is even, then the optimal set regarding the cumulative loss is $\mathcal X^*=\mathcal X$. Therefore, the decision sequence $\\{x_t=1\mid t=1,...,T\\}$ is already optimal. It is easy to check that $R^{alt}=0$ regarding $\{x_t\}$, which is desirable. However, for $R^{psg}$, we have that $R^{psg}=\sup_{x^\*\in[-2,2]}\sum_{1\leq k\leq T/2}(\max\\{1-x^\*, 0\\}+\max\\{x^\*-1, 0\\})=3T$, which is linear w.r.t. $T$.
>
> Given that $R^{psg}$ may produce linear regret even for optimal decisions, we choose to use the alternative form $R^{alt}$ instead. You can also refer to Appendix C for a more detailed explanation and justification of $R^{alt}$.
>
>
> **Q7. "the theoretical results do not have a strong connection with multi-task DNN (in OCO framework, the functions are convex)"**
>
> Our theoretical analysis indeed has a strong connection with multi-task DNNs. Here we explain the reasons from two aspects.
>
> From the *algorithmic* aspect, recall that MGDA is commonly used to train multi-task DNNs, yielding promising empirical performance [6]. Inspired by our theoretical analysis, we designed a new algorithm DR-OMMD as an improvement of MGDA. Empirically, we show that DR-OMMD indeed achieves a better performance in training multi-task DNNs (see Appendix F.3 for more details).
>
> From the *theoretical* aspect, recall that most of the commonly used optimizers for training DNNs, such as AdaGrad [7], Adam [8], and AMSGrad [9], are developed and analyzed in the OCO setting. However, regarding multi-objective optimizers, currently, there is no analysis in the OCO setting. Our proposed multi-objective OCO framework lays a solid foundation for analyzing related multi-task DNN optimizers.
>
> As the first work to give a systematic study of multi-objective OCO, we feel that our work is already self-contained. The analysis in the non-convex setting can be left as future work.
>
> **Reference**
>
> [6] Sener and Koltun. Multi-task learning as multi-objective optimization. 2018.
>
> [7] Duchi et al. Adaptive subgradient methods for online learning and stochastic optimization. 2011.
>
> [8] Kingma and Ba. Adam: a method for stochastic optimization. 2015.
>
> [9] Reddi. On the convergence of Adam and beyond. 2019.

---

> > ### Comment · Reviewer_aTnA · 2022-08-07
> > **Some reservations post-rebuttal**
> >
> > Thank you for the response! The rebuttal has partially addressed my concerns, yet I still have some reservations about the paper on the feasibility of step size setting and the clarity on the paper's motivation.
> > - As discussed in Q4, the method in this paper requires $V_T$ as an input to set step size. The authors assume the availability of $V_T$ at the beginning time stamp (both in analysis and experiments). But, $V_T$ is related to the gradually observed loss function $f_t$ and is hard to be known in practice. So, to me, I still find the method somewhat less appealing from a practical aspect. I am also confused about if the grid search can be applied to this method, given that both step size $\eta$ and parameter $\alpha$ are both related to $V_T$.
> > - Another concern is about the paper's motivation. In lines 43-45 (updated version), one key motivation is to present theoretical foundations of developing a new optimizer for muti-task DNN. Some works including AdaGrad, Adam and AMsGrad are listed as examples to motivate the paper. The listed works are related because the final goal of DNN training is to obtain a single model, and static regret is provided. This paper provides rigorous guarantees on PSG in dynamic regret cases, but its relationship to the paper's initial motivation is not well discussed. Perhaps, a more proper measure is static regret. However, it is still a little bit vague to me how is the proposed alternative static regret related to the goal.
> >
> > As mentioned in my review, I agree that the conversion to single-objective online optimization  (that is, learning weight $\lambda_t$ by another online procedure and then plugging back to OMD) is interesting. But, I am still concerned about the practical implementation and the clarity of the paper's motivation. I believe addressing those issues and also incorporating the modifications promised in the rebuttal will make the paper highly competitive.

---

> > > ### Author Response · Authors · 2022-08-08
> > > **Author Response to Your Reservations**
> > >
> > > Thank you for your additional feedback!
> > >
> > > **1. Regarding your reservation about "the feasibility of step size setting"**
> > >
> > > In fact, in online learning, it is ubiquitous that the choice of hyperparameters depends on the value of some parameters that are hard to know in advance [1, 3]. Typically, there are many approaches to handle unknown parameters. Here we discuss three of them in the existing literature.
> > >
> > > The first way is to conduct a grid search on the unknown parameter [2]. Similar to the grid search conducted in [2], we can directly perform a grid search on $V_T$ and then derive $\eta$ and $\alpha$ from $V_T$.  Specifically, we set the grid of $V_T$ as $\\{2^i\mid 1\leq i\leq N_1\\}$. Then the hyperparameters can be set as $\eta=\frac{2}{G}(\frac{\gamma D v}{GT})^{1/3}$ and $\alpha=\frac{8FG^2T}{v}$ for $v\in\\{2^i\mid 1\leq i\leq N_1\\}$. Following [2], we choose the best setting from the above grid. In the simulation experiment, the best value of $v$ is 13. The average regret of DR-OMMD using grid search is 0.0211. It is slightly worse than DR-OMMD using the true $V_T$ (0.0209) but still substantially better than the *min-norm* (0.0772) and *lin-opt* (0.0302) baselines.
> > >
> > > The second way is to assume $V_T=T^\beta$ and estimate the value of $\beta$ [3]. In the simulation experiment, we estimate $\beta$ according to the first few (say 1000) rounds and derive $\hat\beta=0.95$. Then, we can use $V_T=T^{0.95}$ to decide the hyperparameters $\eta$ and $\alpha$. Empirically, DR-OMMD using this method attains an average dynamic regret of 0.0216.
> > >
> > > The third way is to combine the grid search with a meta-algorithm [4], as you have suggested. In practice, we can run multiple DR-OMMDs where $\eta$ and $\alpha$ are decided by the above grid $V_T\in\\{2^i\mid 1\leq i\leq N_1\\}$, and apply the VariationHedge method [4].
> > >
> > > **2. Regarding your reservation about "the clarity on the paper's motivation"**
> > >
> > > **Q2-1. "This paper provides rigorous guarantees on PSG in dynamic regret cases, but its relationship to the paper's initial motivation is not well discussed. "**
> > >
> > > In fact, as we have discussed in the introduction section, our paper has two initial motivations. The first motivation is to make in-time predictions on streaming data, which motivates both the dynamic regret and the static regret. The second motivation is to lay a theoretical foundation for analyzing multi-task optimizers, which motivates the static regret.
> > >
> > > **Q2-2. "Perhaps, a more proper measure is static regret. However, it is still a little bit vague to me how is the proposed alternative static regret related to the goal."**
> > >
> > > Actually, we have discussed the physical meaning of $R_{MOS}(T)$. The discussion has been put in Appendix C due to the lack of space. Specifically, in Proposition C.1, we show that $R_{MOS}(T)$ measures PSG from the actual cumulative loss $\sum^T_{t=1}F_t(x_t)$ to the Pareto front of the cumulative loss function $\mathcal P^*=\mathcal P_X(\sum^T_{t=1}F_t)$. Hence, minimizing $R_{MOS}(T)$ imposes the learner to generate decisions that reach the Pareto optimal set.
> > >
> > > In addition, as discussed in the preliminary section, as a critical notion of optimality in multi-objective optimization, Pareto optimality is commonly used as the goal of multi-task DNN optimizers such as MGDA and CAGrad. Therefore, as a Pareto optimality metric in the OCO setting, our proposed regret is closely related to the goal of multi-task deep learning.
> > >
> > > **Reference**
> > >
> > > [1] Rakhlin and Sridharan. Online learning with predictable sequences. 2013.
> > >
> > > [2] McMahan and Streeter. Delay-tolerant algorithms for asynchronous distributed online learning. 2014.
> > >
> > > [3] Besbes. Non-stationary Stochastic Optimization. 2015.
> > >
> > > [4] Zhao et al. Dynamic regret of convex and smooth functions. 2020.

---

> > > > ### Comment · Reviewer_aTnA · 2022-08-10
> > > > **Reviewer response: about the step size issue**
> > > >
> > > > Thank you for the second round of feedback and the detailed explanations! My concerns on the motivation (especially on the DNN part), comparison with the previous work, and the technical contribution are properly addressed by the rebuttal.
> > > >
> > > > As for the step size tuning issue, I am not still quite sure whether the listed method can lead to a provable dynamic regret bound. For example, the first method attempts to set the step size by "choosing the best setting from the above grid". However, it is still unclear to me how to select the best setting before all the loss functions are observed. Besides, in the second method, the parameter $\beta$ learned in the first 1000 rounds is used in the latter online learning process. But, in the OCO framework, the latter online functions can be arbitrarily different from the first 1000 rounds. So, it seems hard to provide a guarantee for such a method. As for the third method, I am not sure whether the grid search + meta-algorithm can be directly applied since another online learning process is involved, and there are two parameters $\alpha$ and $\eta$ to be decided.
> > > >
> > > > I believe that there could be many possible ways to set the step size empirically. But, as a theoretical-oriented paper, a feasible algorithm with a provable guarantee would be much more appealing. Overall, I admit this paper has its interesting part on the conversion. But, the requirement of the oracle information of $V_T$ and the lack of a fesible step size setting scheme with a provable guarantee make it less exciting.

---

> ### Author Response · Authors · 2022-08-01
> **Author Response (Part 2)**
>
> Now, we begin to address your questions.
>
> **Q1. "The title of this paper is somewhat overclaimed"**
>
> Actually, it is pretty common to use "online learning" as the title of papers on online convex optimization (see [3, 4] for examples). Admittedly, multi-objective multi-armed bandits have been studied previously, which have been reviewed in Appendix A. To distinguish our work from multi-objective bandits, we can use a more specific title "Multi-objective online convex optimization." Additionally, we have added your mentioned papers to the related work section.
>
> **Q2. "A simple algorithm can achieve $O(V_T)$ dynamic regret. It would be nice to provide a lower bound"**
>
> Our work mainly focuses on the setting where the algorithm only requires the first-order (e.g., gradient) information, which is largely different from your suggested setting where the optimum of $f_t$ can be precisely calculated. Hence, the tightness of our bound should not be evaluated by your mentioned reference. In fact, we have already analyzed the lower bounds of gradient-based methods in the multi-objective OCO setting. The analysis shows that the static regret bound of DR-OMMD matches the lower bound in the single-objective setting, and its dynamic regret bound aligns with the currently best attainable single-objective bound [5]. Due to the lack of space, we have put these discussions in Appendix E.1.
>
> **Q3. "highlight the technical contributions"**
>
> Here we summarize the technical contributions from three aspects. In *framework* formulation, since the vanilla zero-order PSG-based regret is intrinsically complex to be optimized directly via gradient-based methods, we derive its equivalent first-order form via a highly non-trivial transformation, which is very intuitive to the subsequent algorithmic design and theoretical analysis. In *algorithm* design, we motivate it by constructing a non-trivial example of MGDA, showing that it will incur linear regret in the online setting. Inspired by the counterexample, we devise a novel algorithm with double regularization. In *theoretical* analysis, we conduct the first analysis of multiple gradient algorithms in the OCO setting and analyze the tightness of the derived bounds (see Appendix E.1 for more details).
>
> **Q4. "How to choose the parameter $V_T$ in the experiments? Previous works solve this issue by grid search with a meta-algorithm. How does the algorithm in this paper handle the non-stationarity?"**
>
> In the simulation experiment (in Section 5.1), since the whole setting is manually designed, we can directly calculate $V_T$ in advance. Note that our real-world experiments (in Section 5.2 and Appendix F) are conducted in the static setting, which is unrelated to $V_T$.
>
> Surely, our proposed method can be combined with your suggested meta-algorithms by using it as the inner loop. Our paper did not cover these advanced dynamic methods since we aim to develop the first general framework, algorithm, and analysis of multi-objective OCO. Of course, based on our work, more advanced techniques can be designed in the future. We have added a related discussion in the revision.
>
> Recall that our proposed DR-OMMD is based on OMD. At the current stage, as a dynamic algorithm, OMD already handles the non-stationarity to some extent and attains sublinear dynamic regret [5]. As you suggested, more advanced dynamic methods can still be combined with min-regularized-norm. We believe this will be left as interesting future work.
>
> **Reference**
>
> [3] Hazan. Projection-free online learning. 2012.
>
> [4] Duchi et al. Adaptive subgradient methods for online learning and stochastic optimization. 2011.
>
> [5] Besbes. Non-stationary stochastic optimization. 2015.

---

> ### Author Response · Authors · 2022-08-03
> **Author Response (Part 1)**
>
> Thank you for your detailed review!
>
> After reading your summary, we respectfully feel that you may have misunderstood some parts of our paper, i.e., simply regarded it as "*convert the regret minimization problem to a single-objective optimization problem, then solve it using the online mirror descent method.*" In the following, we would like first to clarify these possibly misunderstood points.
>
> **1. The regret minimization problem over PSG is NOT converted to a single-objective optimization problem**
>
> Recall that we have used a novel reduction to convert the regret based on PSG to the following equivalent form:
> $$R\_{MOD}(T)={\rm sup}\_{x^\*_t\in\mathcal X^\*_t, 1\leq t\leq T}{\rm inf}\_{\lambda^\*_t\in\Delta_m, 1\leq t\leq T}\sum\_{t=1}^T({\lambda^\*_t}^\top F_t(x_t)-{\lambda^\*_t}^\top F_t(x^\*_t)).$$
>
> Admittedly, at first glance, minimizing the derived equivalent form looks like a single-objective optimization problem over ${\lambda^*_t}^\top F_t(x)$ if $\lambda^*_t$ is known in advance. However, since $\lambda_t^*$ needs to be inferred from the infimum that depends on the unknown $x^*_t$, $\lambda^*_t$ is also unknown. To cope with this difficulty, in our algorithmic design, we propose a new solver to generate appropriate $\lambda_t$ instead. Obviously, the generation of $\lambda_t$ is not in the realm of single-objective optimization.
>
> **2. The proposed algorithm is NOT vanilla online mirror descent**
>
> In fact, in multi-objective optimization, the core module of most multiple gradient methods, such as MGDA [1] and CAGrad [2], is the computation of composite weights, which determine the composite gradient. The derived composite gradient can be applied to any single-objective optimizer for the model update.
>
> *2-1. Vanilla min-norm will incur linear regret*
>
> To generate the composite weights $\lambda_t$, the most direct way is to apply the min-norm solver commonly used in offline multi-objective optimization. However, in this work, we show that directly applying min-norm is not workable in the online setting. Specifically, we construct a non-trivial example in which the composite gradient generated by min-norm will reversely optimize the loss at each round, incurring a linear regret. See more details in Section 4.1.
>
> *2-2. A meaningful MO-OCO algorithm needs double regularization*
>
> The counterexample shows that only regularizing the iterate $x_t$, as in OMD, is not enough to guarantee sublinear regrets in the multi-objective online setting. To fix this issue, we devise a novel min-regularized-norm solver with an explicit regularization on the composite weights. Equipping it with OMD results in our proposed algorithm. In fact, double regularization is the main algorithmic novelty of this work. See more details in Section 4.2.
>
> *In summary*, DR-OMMD is a novel method well tailored to the multi-objective OCO setting with new algorithmic designs. Notably, it is **NOT** simply a translation to the single-objective setting. You seem to underestimate its novelty, presumably because you might have missed some parts in the whole derivation process of our algorithm. We hope that our explanation can help you better understand it. If you have any additional questions, we are readily prepared to address them in the discussion phase.
>
> **Reference**
>
> [1] Sener and Koltun. Multi-task learning as multi-objective optimization. 2018.
>
> [2] Liu et al. Conflict-averse gradient descent for multi-task learning. 2021.

---

### Official Review · Reviewer_5pLZ · 2022-07-10

**Rating:** 5
**Confidence:** 3
**Soundness:** 4 excellent
**Presentation:** 3 good
**Contribution:** 3 good

**Summary:**

The paper studies multi-objective online learning. It proposes new notions of static regret and dynamic regret based on a Pareto suboptimality gap metric. They show that OMD with the min-norm solver has linear regret, motivating the design of new algorithms. They propose a novel algorithm, Doubly regularized Online Mirror Multiple Descent, that regularizes the composite weights. They show that it obtains static and dynamic regret bounds with the usual scalings in T and that match the optimal bounds when there is only one objective. They provide experiments comparing their algorithm to a linearization baseline and the OMD algorithm with the min-norm solver.

**Questions:**

What is the the definition of regret in figure 1: static or regret? This is important for interpreting the experimental results.

In several places, the authors mention the optimal weights (e.g., "...lin-2 uses thwe optimal lambda_0 for the last..." line 333 and the caption in Figure 2). What is meant by the optimal weight?

I think explaining the improvements over the linearization baseline is very important.

**Limitations:**

Yes.

**Strengths And Weaknesses:**

Strengths:

The paper is clearly written and provides a nice pedagogical overview of the field.

Multi-objective online learning is an important problem with many real-world applications. The new algorithms could find widespread use.

The paper develops seemingly new regret notions with new algorithms and provides experiments showing that their method is competitive.

It is nice that they give a simple example showing that MGDA doesn't work in the multi-objective setting. This is useful motivation for the design of new algorithms.

Weaknesses:

The theoretical comparison to the linearization baseline seems incomplete. It would be nice to understand what a standard analysis for linearization baseline yields for its regret bounds and how this may be suboptimal in comparison to the proposed algorithm. For example, it seems that in the static regret setting removing the inf in the display between lines 210 and 211 and applying the standard analysis also gives a $O(\sqrt{T})$ regret. Is that correct? In lines 299-301, there is some discussion of the differences, but it is unclear how large of a difference there is between the methods. Are there concrete examples as for MGDA?

In the experiments, although the proposed method performs strongly, the improvement over the linearization baseline does not seem significant. This is especially important since the proposed method is more computationally burdensome than the linearization baseline, requiring to solve the optimization problem in in lines 266-267. This makes it more important to show an improvement over linearization.

It seems like an attractive property of MGDA is that it balances between the objectives (lines 151-153). Is there a similar property for this work?

In Section 5.2, it is not clear to me what is the advantage of choosing the regularization strength in an adaptive way. Won't this just yield one point on Pareto Front, so it is not significant? Can you describe the advantage in more detail?

While the upper bounds have the attractive property of achieving the optimal rates in the single-objective setting, the authors give no lower bounds for the multi-objective setting, so optimality of the approach is unclear. Can the authors comment on optimality?

---

> ### Author Response · Authors · 2022-08-01
> **Author Response (Part 2)**
>
> **Q4. "What is the advantage of choosing the regularization strength in an adaptive way?"**
>
> Traditional regularized online learning methods directly add a regularization term to the original loss [6]. This formalism will inevitably encounter the conflicting issue [7]. Specifically, since regularization aims to avoid overfitting, it often conflicts with the optimization of the original loss. As a consequence, the original loss may not be sufficiently optimized. Our adaptive regularization experiment provides a new perspective on regularized online learning. It views regularization as one objective beyond the loss objective. Using multiple gradient methods, we can adaptively select the regularization strength near the common descent direction, which can alleviate the conflict issue and optimize the primary loss more sufficiently.
>
> We also note that in this experiment, our goal is not simply reaching an arbitrary point on the Pareto front. Recall that the ultimate goal of adding regularization is to handle overfitting and improve the primary goal. Hence, the regularization strength needs to be appropriately set. Compared to linearization using a fixed regularization strength, our adaptive approach brings more flexibility and empirically appears more robust to the choice of regularization strength.
>
> **Q5. The optimality of DR-OMMD**
>
> Actually, in Appendix E.1, we have analyzed the lower bound of gradient-based methods in the multi-objective online setting, showing that DR-OMMD's regret bound is w.r.t. $T$ and $m$. Due to the lack of space, we have put it in the appendix.
>
> **Q6. "What is the definition of regret in Figure 1?"**
>
> In simulation experiments, we present the dynamic regret. Note that we adopt the static regret in all real-world experiments (Section 5.2 and Appendix F).
>
> **Q7. "What is meant by the optimal weight?"**
>
> Recall that our experiments conduct a grid search on the linearization weights for the linearization baseline. In simulation experiments, we measure the average regret. Hence, the optimal weights refer to the linearization weights that attain the lowest cumulative regret within certain rounds (e.g., the first $T_1$ rounds for lin-1, and the last $T_2$ rounds for lin-2). In adaptive regularization experiments, we intend to introduce appropriate regularization to improve the primary goal. Hence, the optimal weights refer to the linearization weights that incur the lowest overall classification loss.
>
> **Reference**
>
> [6] Xiao. Dual averaging method for regularized stochastic learning and online optimization. 2009.
>
> [7] Sener and Koltun. Multi-task learning as multi-objective optimization. 2018.

---

> ### Author Response · Authors · 2022-08-02
> **Author Response (Part 1)**
>
> Thank you for your insightful review!
>
> **Q1. Theoretical comparison to linearization. Are there concrete examples as for MGDA?**
>
> Surely, the bound of linearization is also in the order of $O(\sqrt T)$. In Appendix E.2, we have detailedly discussed this point. Note that we have put it to the appendix due to the lack of space. Of course, minimizing linearization can also reach the Pareto front. This is not a problem of our framework, but actually an intrinsic requirement of Pareto optimality, which can be achieved in a way by simply minimizing any linearization (see Proposition 8 in [1] for detailed proofs). As a general framework, our framework (especially the regret) should incorporate this particular case. Hence, it is unsurprising that linearization also achieves $O(\sqrt T)$ regret.
>
> In this paper, we have shown that our regret is indeed smaller than linearization with fixed weights. This is the best result we can get for now. Empirically, we observe that our proposed method consistently outperforms linearization. Given the promising empirical results, we believe that it is possible to design better algorithms with explicitly better regret bounds. Admittedly, it is sometimes difficult to strictly prove the advantage of an optimizer theoretically. A very typical case is Adam [4] versus AdaGrad. Since both algorithms acheve $O(\sqrt T)$ regret, Adam does not show obvious theoretical superiority over AdaGrad. Its advantage is supported by extensive empirical results.
>
> As for MGDA, in the offline setting, previous studies only show that it has the same convergence rate as linearization [2, 3]. Its advantage over linearization is from the algorithmic aspect. Specifically, MGDA uses a common descent direction to alleviate the conflicting gradients [5], thus empirically outperforms linearization.
>
> **Q2. "The improvement over linearization does not seem significant... more computationally burdensome"**
>
> Actually, in our real-world experiments, DR-OMMD has more than $5\\%$ decrease in the average loss compared to linearization (see Section 5.2 and Appendix F). This can be considered significant, given that the improvement of MGDA over linearization in the stochastic setting is less than $2\\%$ [5]. Besides, when $m=2$ (which is one of the most frequently used scenarios in MOO), min-regularized-norm incurs negligible computational burdensome, since it has a closed-form solution.
>
> **Q3. "MGDA balances between the objectives. Is there a similar property for this work?"**
>
> Our work is built upon MGDA, which uses a common descent direction to balance between different objectives. However, in Section 4.1, we have proven that in the online setting, purely pursuing common descent like MGDA will incur linear regret. To overcome the non-convergence problem, we have to additionally introduce some constraint to the stability of the composition weights. In this paper, we propose to achieve this by introducing a regularizer to the original min-norm solver, which results in the doubly regularized algorithm. This algorithm achieves a compromise between the common descent property and the stability of the composition weights. Hence, our algorithm also inherits the common descent property from MGDA, though a bit weaker than MGDA.
>
> **Reference**
>
> [1] Emmerich. A tutorial on multi-objective optimization. 2018.
>
> [2] Fliege et al. Complexity of gradient descent for multi-objective optimization. 2019.
>
> [3] Liu et al. Conflict-averse gradient descent for multi-task learning. 2021.
>
> [4] Kingma and Ba. Adam: a method for stochastic optimization. 2015.
>
> [5] Sener and Koltun. Multi-task learning as multi-objective optimization. 2018.

---

### Official Review · Reviewer_yhTW · 2022-07-11

**Rating:** 6
**Confidence:** 4
**Soundness:** 3 good
**Presentation:** 3 good
**Contribution:** 3 good

**Summary:**

This paper focuses on the problem of multi-objective online convex optimization, where the losses are given in vectors and the scalarized objective is the Pareto suboptimality gap, which to some extent reflects the distance to the Pareto frontier. The authors proposed online mirror multiple descent, which is the generalized version of online multiple gradient descent by allowing different regularizers, and provably enjoy optimal dynamic regret. Experiments validate the effectiveness.

**Questions:**

Does the term the $\sum_t \lambda_t-\lambda_0$ in Thm 4.6 have some specific meaning?

**Limitations:**

See weaknesses above.

**Strengths And Weaknesses:**

Strengths:
- Clear writing
- Standard techniques with strong results

Minor Weaknesses:
- There exists mainly two lines of work on multi-objective online learning. One line focuses on the Pareto frontier while another line focuses on deriving no-regret algorithms according to Blackwell's approachability theorem. Though slightly different in the explicit scalarized objective, I think both lines of work contribute a lot to the community. And thus some additional related works and contenders should be noticed.
- The implementation of projection with different types of regularizers in OMMD should be carefully discussed.

---

> ### Author Response · Authors · 2022-08-01
> **Author Response**
>
> Thank you for your constructive review!
>
> **Q1. "Some additional related works and contenders should be noticed"**
>
> We have given a detailed literature review. Due to the lack of space, we have put it to Appendix A. It includes your mentioned papers on Blackwell's approachability and also other related works such as Pareto frontier in multi-objective bandits and multi-objective stochastic optimization. We have also added one more reference on Pareto frontier, which is suggested by Reviewer aTnA.
>
> **Q2. "The implementation of projection with different types of regularizers in OMMD should be carefully discussed"**
>
> There are actually two regularizers in DR-OMMD: one regularizing the iterates $x_t$ and the other regularizing the composition weights $\lambda_t$. The former regularizer is exactly the original regularizer in single-objective OMD, whose calculation can be found in many literatures on OCO [1, 2]. For example, when using L2-regularization, OMD recovers the standard online gradient descent algorithm. For more details, please refer to the textbook (Section 5.4 in [2]).
>
> The latter regularizer on $\lambda_t$ is a novel characteristic of MO-OCO. In the following, we discuss the implementation of different types of regularizers on $\lambda_t$.  Analogous to the discussion on the former regularizer [1], we mainly consider L1-regularization and L2-regularization. Recall that in Section 4.2 we have presented the closed-form solution to L1-regularization when $m=2$; the detailed derivation and the calculation for $m>2$ are deferred to Appendix D.1 due to the lack of space.
>
> Here we discuss the implementation of min-regularized-norm with **L2-regularization** $\frac{1}{2}\Vert\lambda-\lambda_0\Vert^2_2$, which takes the following form
> $$\lambda_t={\rm argmin}_{\lambda\in\Delta_m}\Vert\nabla F_t(x_t)\lambda\Vert^2_2+\frac{\alpha}{2}\Vert\lambda-\lambda_0\Vert^2_2.$$
>
> When $m=2$, it has a closed-form solution. Denote the gradients $g_1=\nabla f^1_t(x_t), g_2=\nabla f^2_t(x_t)$, then the closed-form solution is given as $\lambda_t=(\gamma_t^1,1-\gamma_t^1)$ where
> $$\gamma^1_t=\max \\{ \min \\{\frac{(g_{2}-g_1)^\top g_2+\alpha\gamma_0^1}{\Vert g_2-g_1\Vert^2+\alpha}, 1\\}, 0 \\}.$$
>
> When $m>2$, since $\lambda_t$ is constrained to the probability simplex $\Delta_m$, we can use a Frank-Wolfe method to efficiently calculate the composition weights. The derivation is analogous to the implementation of L1-regularization. Specifically, at each round $t$, denote the gradients $g_i=\nabla f^i_t(x_t), i\in\\{1,...,m\\}$ and $G=\nabla F_t(x_t)$, we initialize $\lambda_t=(\gamma^1_t,...,\gamma^m_t)=(\frac{1}{m},...,\frac{1}{m})$ and loop over:
>
> 1. Select an index $k\in{\rm argmax}\_{i\in\\{1,...,m\\}}\{\sum^m_{j=1}\gamma_t^i g_i^\top g_j+\alpha(\gamma_t^i-\gamma_0^i)\}$;
>
> 2. Compute $\delta\in{\rm argmin}\_{0\leq\delta\leq1}\Vert\delta g_k+(1-\delta)G\lambda_t\Vert^2_2+\frac{\alpha}{2}\Vert\delta(e_k-\lambda_t)+\lambda_t-\lambda_0\Vert^2_2$, which has an analytical form
>     $$\delta=\max\\{\min\\{\frac{2(G\lambda_t-g_k)^\top G\lambda_t+\alpha(\lambda_t-e_k)^\top(\lambda_t-\lambda_0)}{2\Vert G\lambda_t-g_k\Vert^2+\alpha \Vert \lambda_t-e_k\Vert^2}, 1\\}, 0\\};$$
>
> 3. Update $\lambda_t=(1-\delta)\lambda_t+\delta e_k$.
>
> Typically this recursion will converge within a few steps [3]. Note that since the line search in step 2 has a closed-form solution, its calculation is very light, which is the same as that of original min-norm [3]. Other types of regularizers can also be derived in a similar way, which can be left as the future work.
>
> **Q3. "Does $\sum_t\Vert\lambda_t-\lambda_0\Vert_1$ have some specific meaning?"**
>
> Mathematically, this term calculates the cumulative difference in L1-norm between $\lambda_t$ and $\lambda_0$. Physically, it measures the stability of composition weights $\lambda_t$. The smaller this term, the more stable $\lambda_t$.
>
> **Reference**
>
> [1] Xiao. Dual averaging method for regularized stochastic learning and online optimization. 2009.
>
> [2] Hazan. Introduction to online convex optimization. 2016.
>
> [3] Sener and Koltun. Multi-task learning as multi-objective optimization. 2018.

---

### Official Review · Reviewer_6wxE · 2022-07-12

**Rating:** 7
**Confidence:** 3
**Soundness:** 3 good
**Presentation:** 4 excellent
**Contribution:** 3 good

**Summary:**

This work performs a systematic study of online multi-objective optimization.  The authors formulate the multi-objective online convex optimization, in which multi-objective dynamic regret and static regret are formulated.
The authors further show that the vanilla min-norm method used in MGDA may incur linear regrets. The authors further proposed a regularized version of online mirror descent. The min-regularization-norm is inspired by the regularization in FTRL. Both static regret bound and static regret bound are derived for the proposed regularized multi-objective online mirror descent method. The regret bounds match the optimal regret bounds in the single-objective setting.



**Questions:**

Can the authors leverage more on what are the meaningful and high-impact applications of online convex optimization?

**Limitations:**

- The empirical evaluation is not comprehensive enough.
- The motivation and significance of the contribution need justification.
- The proposed method is in fact quite straightforward. The novelty is kind of limited.

**Strengths And Weaknesses:**

Strength:
- The multi-objective online convex optimization framework is quite clean and helpful.
- The proposed algorithm is reasonable and sound. The theory is solid to me.
- Good empirical performance.

---

> ### Author Response · Authors · 2022-08-01
> **Author Response**
>
> Thank you for your excellent review!
>
> **Q1. More meaningful and high-impact applications of online convex optimization**
>
> The most meaningful and high-impact applications of OCO are to lay solid theoretical foundations for analyzing different optimizers. Currently, nearly all the optimizers for training DNNs are initially analyzed in OCO, such as AdaGrad [1] and Adam [2]. Besides, the theoretical results based on the OCO framework may inspire the design of new optimizers, such as AdaGrad [1] and AMSGrad [3]. For example, AdaGrad is designed to yield a smaller regret bound associated with gradient norms by adaptively selecting appropriate step sizes; AMSGrad is proposed to fix one convergence proof bug as associated with the change of the inverse of adaptive learning rates in Adam.
>
> As for the multi-objective setting, recently, there has been a surge of research interest in designing multi-objective optimizers, resulting in MGDA [4], CAGrad [5], and PCGrad [6], etc. However, their analyses are mainly conducted in the offline setting rather than the typical OCO setting. This paper fills this gap by developing a general formal framework to analyze multi-objective optimizers in the OCO setting. Meanwhile, our proposed algorithm is primarily inspired by the theoretical analysis, showing consistently improved performance than the previous method. We believe that our MO-OCO framework can also be used to analyze other MOO optimizers and inspire the design of new optimizers for training multi-task DNNs.
>
> **Q2. "The empirical evaluation is not comprehensive enough"**
>
> Actually, we have conducted extensive experiments, which include numerical simulation experiments (Section 5.1), real-world convex experiments (Section 5.2 and Appendix F.2), and non-convex multi-task deep learning experiments (Appendix F.3). Due to the lack of space, we have put some parts of them to the appendix. Presumably, you might have missed some of them. The experiments demonstrate the effectiveness of our proposed algorithm in both convex and non-convex settings.
>
> Note that our experiments also contribute new perspectives to understanding regularization. Specifically, in real-world convex experiments, we view the regularization as one objective besides the original loss objective. Such a new formalism allows us to take a multi-objective approach to this problem. By using our proposed multiple gradient algorithm, we successfully alleviate the conflicts between the regularization and the original loss, which is unavoidable using traditional methods. Moreover, our proposed method can adaptively select the regularization strength, which brings more flexibility and is more robust to the choice of strength.
>
> **Q3. More justification of the motivations**
>
> We have justified the motivations from two aspects. At the *algorithm* level, to motivate the design of the DR-OMMD algorithm, we construct a non-trivial example to show that traditional MGDA will incur linear regret in the online setting. At the *framework* level, the motivation for developing a multi-objective OCO framework is to provide a theoretical foundation for analyzing MOO optimizers (please refer to our response to **Q1**).
>
> **Q4. The significance and novelty of this work**
>
> The significance and novelty of this paper are mainly two folds. In *framework* formulation, since the zero-order PSG-based regret is intrinsically complex to be optimized directly via gradient-based methods, we derive its equivalent unconstrained max-min form via a highly
> non-trivial transformation, which is very intuitive to the subsequent algorithmic design and theoretical analysis. In *algorithm* design, the example of MGDA's incurring linear regret is fundamentally new. Enlightened by this analysis, we propose to use double regularization, i.e., regularizing both the iterates and the composition weights, which is a novel characteristic of multi-objective online learning. Note that L1-regularization is just a straightforward way to regularize the composition weights.
>
> **Reference**
>
> [1] Duchi et al. Adaptive subgradient methods for online learning and stochastic optimization. 2011.
>
> [2] Kingma and Ba. Adam: a method for stochastic optimization. 2015.
>
> [3] Reddi et al. On the convergence of Adam and beyond. 2019.
>
> [4] Sener and Koltun. Multi-task learning as multi-objective optimization. 2018.
>
> [5] Liu et al. Conflict-averse gradient descent for multi-task learning. 2021.
>
> [6] Yu et al. Gradient surgery for multi-task learning. 2020.

---

### Author Response · Authors · 2022-08-04
**General Author Response**

We would like to thank all reviewers for the detailed and constructive reviews. We hope our response addresses the concerns. We are very happy to address any additional questions that may arise during the discussion period.

---

### Meta-Review · Area_Chair_mW3i · 2022-08-26

**Recommendation:** Reject
**Confidence:** Less certain

**Metareview:**

First, I must say that this is quite an interesting and elegant work, and regardless of the decision on this paper, I would like the authors to continue along this path. The paper is nicely written with a good flow, and the authors naturally arrive at their new algorithm, Doubly Regularized Online Mirror Multiple Descent. In the static case, the regret bounds the authors can show are solid and there are no complaints from any of the reviewers. This is almost true for the dynamic regret case, except for the large caveat that (in terms of theory) the authors' algorithm needs to know $V_T$. I discuss this issue more below. From the experiments side, although the authors show some improvement over the linearization baseline, the sense from the reviewers is that this improvement is not that much. For this reason, I truly think this paper needs to be solid theoretically.

**Regarding knowledge of $V_T$:**
First, to make it seem fair to request adaptivity to $V_T$, let me mention that in, e.g., the work of Campolongo and Orabona (2021) — which the authors cited in the context of their assumption of known $V_T$ — there are algorithms that automatically adapt to unknown $V_T$. Now, the authors' current need for knowledge of $V_T$ wouldn't necessarily be a problem if a doubling trick or meta-algorithm could be used. It is quite unclear whether the former could work, and in the reviewer discussion we have doubts about whether VariationHedge could be used in order to get adaptivity to $V_T$. I strongly suggest the authors look into this. Also, I think it is worth mentioning that the authors mentioned in one of their responses to a reviewer that their algorithm only uses first-order (gradient) feedback. It seems that this would mean that, even at the end of the game, $V_T$ could not be computed.

In light of the currently unclear significant benefit over linearization empirically and also the issue with adapting to $V_T$ for the dynamic regret results, this paper does not quite meet the bar for acceptance. However, the decision was a close one, and I strongly encourage the authors to increase discussion of both issues (and technically address adaptivity to $V_T$, working out details using VariationHedge if the authors can indeed do this).

**Award:**

No

---

### Decision · Program_Chairs · 2022-09-14

Reject